# Cluster-Based Memetic Approach of Image Alignment

**Cătălina Lucia COCIANU and Cristian Răzvan USCATU \***

Department of Economic Informatics and Cybernetics, Bucharest University of Economic Studies, 10552 Bucureşti, Romania; catalina.cocianu@ie.ase.ro

\* Correspondence: cristian.uscatu@ie.ase.ro

**Abstract:** The paper presents a new memetic, cluster-based methodology for image registration in case of geometric perturbation model involving translation, rotation and scaling. The methodology consists of two stages. First, using the sets of the object pixels belonging to the target image and to the sensed image respectively, the boundaries of the search space are computed. Next, the registration mechanism residing in a hybridization between a version of firefly population-based search procedure and the two membered evolutionary strategy computed on clustered data is applied. In addition, a procedure designed to deal with the premature convergence problem is embedded. The fitness to be maximized by the memetic algorithm is defined by the Dice coefficient, a function implemented to evaluate the similarity between pairs of binary images. The proposed methodology is applied on both binary and monochrome images. In case of monochrome images, a preprocessing step aiming the binarization of the inputs is considered before the registration. The quality of the proposed approach is measured in terms of accuracy and efficiency. The success rate based on Dice coefficient, normalized mutual information measures, and signal-to-noise ratio are used to establish the accuracy of the obtained algorithm, while the efficiency is evaluated by the run time function.

**Keywords:** bio-inspired computing; evolutionary strategies; firefly algorithm; meta-heuristics; rigid transformation; image registration; memetic algorithms; cluster-based methodology

## 1. Introduction

Often inspiration comes from nature and this extends into the field of computer science. Many algorithms mimic the behavior of biological organisms to solve problems that are difficult or impossible to solve in other way. Such algorithms are increasingly used to approach various problems. They are permanently adapted, modified, combined, developed and seem to have a bright future [1].

With the advancement of technology came an avalanche of images that are used in various sectors. More than often, they are images of the same object but they are not identical, being recorded by different sensors, at different times, angles, luminosity and other variations. These images must be processed in order to be used and the huge number of images makes this a perfect candidate for automation. The process of image registration has raised a lot of attention in the last two decades, reflected in numerous papers published. Due to the wide range of variations, many authors turn to bio-inspired evolutionary algorithms. Advancements are regularly surveyed and reported in scientific publications such as [2–6].

Image registration can be applied to both deformable images and rigid transformations and both types are studied through the use of bio-inspired evolutionary algorithms. In [7], an evolutionary algorithm (EA) with multiple objective optimization is used to find the best way to automate registration of deformable images in the medical field. The results indicate the algorithm is suitable for solving problems with limited

deformations and can create better images to be used by experts, also freeing their time by automating image processing.

Registration of images with rigid transformations is advanced in [8] where authors propose a genetic algorithm (GA) that computes best parameters (for translation, rotation and scale) based on matching shapes of molecules. The results have been validated by applying the algorithm for registration of various medical image types: magnetic resonance image (MRI), computed tomography (CT) and positron emission tomography (PET). Images used for testing were obtained from the retrospective image registration evaluation (RIRE) project. The accuracy of the registration is enhanced by using a better fitness function for the genetic algorithm. Authors highlight the fact that the most commonly used similarity function (mutual information of two images) has local optimums which bring the risk of the algorithm becoming stuck in one such point. Improvements are needed in order to ensure the global optimum is found and for this purpose the authors combine the widely-used mutual information function with an interaction energy function.

All kinds of bio-inspired algorithms are used in reported works, from GA [8, 9] and evolutionary algorithms (EA) [7] to newer approaches that employ hybridizations and metaheuristics.

Various articles report on the use of swarm intelligence and derivate algorithms for image registration. An in-depth study of particle swarm optimization, with shortcomings and numerous developments and hybridizations is reported in [10]. In [11], authors also review such algorithms and hybridization with evolutionary strategies (ES) for biological and medical image registration, indicating promising results for future developments.

The GA approach is compared to the artificial bee colony (ABC) approach in [9], highlighting the advantages of each algorithm: while GA is faster, ABC gives better quality of image registration. Another comparison between GA and swarm approach, using the correlation function of two images to estimate the quality of registration process, is reported in [12] with the conclusion that the PSO approach provides superior results.

Particle swarm optimization sample consensus (PSOSAC) is used in [13] to optimize registration efficiency. The results are compared against random sample consensus (RANSAC) algorithm and proved to lead to better results.

In [14] authors use an adaptation of coral reef optimization algorithm with substrate layers (CRO-SL) with real numbers for encoding the information. Both feature-based and intensity-based variants for registration are attempted. This approach is compared with others and yields very good results.

Bacterial foraging optimization (BFO) algorithm is applied on image registration in [15,16]. Results are compared with those of other recent algorithms proving to be competitive.

The intensive calculations required for the convergence of bio-inspired algorithms might lead to unfeasible computation time, which is why one of the main concerns is speeding up the algorithms. Use of multiple clusters of data in order to speed up the registration algorithms is an idea presented in several articles. In [17], multiple swarms of ABC are used to this purpose, with very good, reported results regarding the computation time. In [18], authors compare PSO with multi-swarm optimization (MSO) and cuckoo search algorithm (CSA) for image registration. For the dataset, used PSO offers the best precision, while PSO and MSO offer best speed and CSA and MSO offer the least scatter of results. As such, no algorithm prevails on all criteria, but authors mention that these results might be particular to the problem solved and may be different in other cases.

The Firefly paradigm is also used to approach the problem of image registration, with results reported in scientific publications. The many local optimums are a trap for algorithms that use mutual information as fitness indicator for image registration, as mentioned before. In [19], Firefly is used to overcome this problem by combining the use of lower and higher resolution variants of an image and the Powell algorithm. Firefly is

used to produce an imprecise result using the lower resolution images, then Powell algorithm is applied on higher resolution images.

A hybrid firefly algorithm (HFA) is used in [20] to solve the problem of slow convergence and for a better coverage of the entire solution space during the search process.

This paper presents a new memetic, cluster-based methodology for image registration. The working assumption is that the sensed images are variants of the targets perturbed by the geometric transformation consisting in rotation, translation and scaling. The proposed approach is applied to align either binary or monochrome images. In both cases, the first step consists in computing the boundaries of the search space based on the object pixels of the processed images. Then, the memetic registration procedure is applied. The alignment of pairs of monochrome images is performed on binarized and resized images. The scaling step is meant to speed up processing of pairs of images and it is used in case of gray scale images only. The quality of the resulted algorithms is measured in terms of accuracy and efficiency. The success rate based on Dice coefficient, the normalized Shannon/Tsallis mutual information measures and signal-to-noise ratio are used to evaluate the accuracy, while the efficiency is established by the run time function. A comparative analysis against two of the most commonly used methods to align images in case of rigid/affine perturbation, namely one plus one evolutionary optimizer [21] and principal axes transform (PAT) [22] experimentally proves the quality of the proposed methodology.

The rest of the paper is organized as follows. The similarity measures used both to define the fitness function and to evaluate the accuracy of the alignment are supplied in Section 2. The proposed methodology is exposed in the core section of the paper. We describe the accuracy and efficiency indices in Section 4. A series of experimental results and the comparative analysis concerning the accuracy and the efficiency of the resulted algorithms are presented next. The final part of the paper includes conclusions and suggestions for further developments regarding bio-inspired methods for image registration.

## 2. Similarity Measures

Let X and Y be two binary sets. The Dice coefficient measures the similarities between X and Y by:

$$\text{Dice}(X, Y) = \frac{2 \cdot |X \cap Y|}{|X| + |Y|} \tag{1}$$

where $|X|$ stands for the cardinal of X. Obviously, $\max\limits_{X, Y \text{ binary sets}} \text{Dice}(X, Y) = 1$ and $\text{Dice}(X, Y) = 1$ if and only if $X = Y$. The Dice coefficient can be directly applied to pairs of binary images.

In general cases of monochrome and colored images, more complex functions should be considered instead, one of the most commonly used being the normalized mutual information computed using entropic measures.

Let X and Y be monochrome images with distributions $p(x)$ and $p(y)$, respectively. Note that $p(x)$ is the probability of intensity x appearing in image X. We denote by $p(x, y)$ the joint probability, that is the probability that corresponding pixels in X and Y have intensity x and y, respectively. The joint probability distribution of the images X and Y reflects the relationship between intensities in X and Y. Assuming that L is the number of grey levels of the images, the Shannon entropy of X is defined by:

$$H^S(X) = -\sum_{x=0}^{L-1} p(x) \cdot \log_2 p(x) \tag{2}$$

The joint Shannon entropy is given by:

$$H^S(X,Y) = -\sum_{x=0}^{L-1}\sum_{y=0}^{L-1} p(x,y)\cdot \log_2 p(x,y) \qquad (3)$$

Shannon normalized mutual information is defined by [23]:

$$NMI^S(X,Y) = \frac{2\cdot MI^S(X,Y)}{H^S(X) + H^S(Y)} \qquad (4)$$

$$MI^S(X,Y) = H^S(X) + H^S(Y) - H^S(X,Y) \qquad (5)$$

where $MI^S(X, Y)$ is the Shannon mutual information. The maximum value of Shannon normalized mutual information is one and it is reached when $X = Y$.

Shannon mutual information is widely used in image registration, but it is sensitive to noise. To reduce the influence of outliers, one can use similarity measures based on Tsallis entropy instead [22,24].

Tsallis entropy of order $\alpha$ is defined by [24]

$$H_\alpha^T(X) = \frac{1}{\alpha - 1}\cdot\left(1 - \sum_{x=0}^{L-1} p(x)^\alpha\right) \qquad (6)$$

The joint Tsallis entropy of order $\alpha$ is given by

$$H_\alpha^T(X,Y) = \frac{1}{\alpha - 1}\cdot\left(1 - \sum_{x=0}^{L-1}\sum_{y=0}^{L-1} p(x,y)^\alpha\right) \qquad (7)$$

Note that when $\alpha$ approaches to 1, Tsallis entropy approaches Shannon entropy.
For $\alpha > 1$, Tsallis mutual information is expressed as:

$$MI_\alpha^T(X,Y) = H_\alpha^T(X) + H_\alpha^T(Y) - H_\alpha^T(X,Y) \qquad (8)$$

and Tsallis normalized mutual information

$$NMI_\alpha^T(X,Y) = \frac{MI_\alpha^T(X,Y)}{H_\alpha^T(X,Y)} \qquad (9)$$

For $\alpha > 1$, the following properties hold [25]
1. $NMI_\alpha^T(X, Y) \in [0,1]$
2. $NMI_\alpha^T(X, Y) = 0$ if X and Y are independent
3. $NMI_\alpha^T(X, Y) = 1$ if $X = Y$.

In our work the fitness function is defined in terms of Dice coefficient, while $NMI^S$ and $NMI_\alpha^T$, $\alpha > 1$, are used to evaluate the accuracy of the registration procedure.

### 3. The Proposed Methodology for Binary Image Alignment

The proposed methodology used to align two binary images is developed to deal with perturbations involving rotation, translation and scaling. Note that in image registration literature, a rigid transformation involves either rotation and translation [4,22], or rotation, translation and scale changes [26]. The first case corresponds to rigid geometric transformations which preserve distances and it is given by three parameters, the translation [a, b] and the rotation angle θ. In the second approach, objects in the images retain their relative shape and position, rigid transformation being defined by four parameters, the translation [a, b], the rotation angle θ and the scale factor s. From the geometrical point of view, the transformation corresponds to a similarity with stretching factor s. In our work, we used the second version of rigid transformations.

Let T be the target image of size M × N. The sensed image S results as a geometric transformation of T, defined in terms of rotation, translation and scale changes. Let θ be the rotation angle defining the rotation matrix $R = \begin{bmatrix} \cos\theta & -\sin\theta \\ \sin\theta & \cos\theta \end{bmatrix}$, s the scale factor and

$[a, b]^T$ the translation vector. For each pixel $(x, y), 1 \leq x \leq M, 1 \leq y \leq N$, the output $S(x, y)$ is given by:

$$S(x, y) = T\left(f(x, y)\right) \tag{10}$$

where

$$f(x, y) = \begin{bmatrix} a \\ b \end{bmatrix} + s \cdot R \cdot \begin{bmatrix} x \\ y \end{bmatrix} \tag{11}$$

The computation of the transformation parameters can be carried out by an evolutionary algorithm, where the boundaries of the search space are established using the sets of object pixels belonging to the target image and to the sensed image respectively.

The components of our methodology are described below.

### 3.1. The Search Space Boundaries

The first stage of our method consists in defining the search space. Based on the assumption that the deformation is reversible, that is each object in T corresponds to a certain object in S, namely its perturbed version, the computation of the search space boundaries is performed taking into account only the object pixels.

We consider that the initial image was rotated to the left, that is $\theta \in [-\pi, 0]$ and the scale factor $s \in (0, smax]$. Obviously, similar results can be obtained in case of right-side rotations. We denote by $\{(x_S^1, y_S^1), \ldots, (x_S^p, y_S^p)\}$ the set of object pixels belonging to S, let $\{(x_T^1, y_T^1), \ldots, (x_T^p, y_T^p)\}$ be the set of object pixels of T and

$$minxS = \min_{i=1,\ldots,p} x_S^i, \ minyS = \min_{i=1,\ldots,p} y_S^i, \ maxxS = \max_{i=1,\ldots,p} x_S^i, \ maxyS = \max_{i=1,\ldots,p} y_S^i \tag{12}$$

$$minxT = \min_{i=1,\ldots,p} x_T^i, \ minyT = \min_{i=1,\ldots,p} y_T^i, \ maxxT = \max_{i=1,\ldots,p} x_T^i, \ maxyT = \max_{i=1,\ldots,p} y_T^i \tag{13}$$

$$maxS = \max\{maxxS, maxyS\} \tag{14}$$

Using (11) we obtain

$$minxT \leq a + s \cdot \cos\theta \cdot x - s \cdot \sin\theta \cdot y \leq maxxT \tag{15}$$

Using straightforward computation, since $\sin\theta \leq 0$, we obtain

$$minxT \leq a + s \cdot (|\cos\theta| \cdot x + |\sin\theta| \cdot y) \leq a + smax \cdot \sqrt{2} \cdot maxS$$

and

$$maxx\, T \geq a - s \cdot |\cos\theta| \cdot x + s \cdot |\sin\theta| \cdot y \geq a - smax \cdot maxxS$$

We obtain the following definition domain of parameter a

$$Da = \left[minxT - smax \cdot \sqrt{2} \cdot maxS, maxxT + smax \cdot maxxS\right] \tag{16}$$

In the same way, since

$$minyT \leq b + s \cdot \sin\theta \cdot x + s \cdot \cos\theta \cdot y = b - s \cdot |\sin\theta| \cdot x + s \cdot \cos\theta \cdot y \leq maxyT \tag{17}$$

we obtain

$$minyT \leq b - s \cdot |\sin\theta| \cdot x + s \cdot |\cos\theta| \cdot y \leq b + smax \cdot maxyS \tag{12}$$

$$maxyT \geq b - s \cdot |\sin\theta| \cdot x - s \cdot |\cos\theta| \cdot y = b - s \cdot (|\cos\theta| \cdot x + |\sin\theta| \cdot y)$$
$$\geq b - smax \cdot \sqrt{2} \cdot maxS \tag{13}$$

and consequently, the definition domain of b is given by

$$Db = \left[minyT - smax \cdot maxyS, maxyT + smax \cdot \sqrt{2} \cdot maxS\right] \tag{20}$$

The proposed image registration method aims to compute the parameter $(a, b, \theta, s)$ such that the relations (10) and (11) hold, where:

$$(a, b, \theta, s) \in \mathcal{D}(S, T) = Da \times Db \times [-\pi, 0] \times (0, smax] \tag{21}$$

### 3.2. Metaheuristics for Image Registration

The binary image registration procedure can be developed using evolutionary approaches. The proposed methodology uses a special tailored version of Firefly algorithm and standard two membered evolutionary strategy (2MES) to compute a solution of (10). In this section, we briefly describe the versions of Firefly algorithm and 2MES specially tailored to binary image registration [27,28].

From the evolutionary algorithms point of view, solving the problem (10) involves defining a search space and a fitness function, and applying an iterative procedure to compute an individual that maximizes the fitness. In our approach, the search space is defined by (19) and, for each candidate solution $c = (ca, cb, c\theta, cs)$, the fitness function measures the similarity between the target image T and the image $\widetilde{T}$,

$$\widetilde{T}(x, y) = S\big(g_c(x, y)\big) \tag{22}$$

$$g_c(x, y) = \frac{1}{cs} \cdot cR^T \cdot \left( \begin{bmatrix} x \\ y \end{bmatrix} - \begin{bmatrix} ca \\ cb \end{bmatrix} \right) \tag{23}$$

$$fitness(c) = Similarity\big(\widetilde{T}, T\big) \tag{24}$$

where $cR = \begin{bmatrix} \cos c\theta & -\sin c\theta \\ \sin c\theta & \cos c\theta \end{bmatrix}$.

Evolutionary Strategies (ES) are self-adaptive methods for continuous parameter optimization. The simplest algorithm belonging to ES class is 2MES, a local search procedure that computes a sequence of candidate solutions based on Gaussian mutation with adaptive step size. Briefly, the search starts with a randomly generated/input vector $c_0$, an initial step size $\sigma_0$ and the values $\vartheta \in [0.817, 1)$ and $\tau$ implementing the self-adaptive Rechenberg rule [29]. At each iteration t, the algorithms computes:

$$c_t = \begin{cases} c_{t-1} + z \,, \text{ if } fitness(c_{t-1} + z) > fitness(c_{t-1}) \\ c_{t-1}, \qquad\qquad\qquad\qquad\qquad \text{otherwise} \end{cases} \tag{25}$$

where $z$ is randomly generated from the distribution $N(0, \sigma_{t-1})$. The dispersion is updated every $\tau$ steps according to Rechenberg rule:

$$\sigma_t = \begin{cases} \dfrac{\sigma_{t-1}}{\vartheta}, & P/\tau > 0.2 \\ \sigma_{t-1} \cdot \vartheta, & P/\tau < 0.2 \\ \sigma_{t-1}, & P/\tau = 0.2 \end{cases} \tag{26}$$

where p is the number of distinct vectors computed by the last $\tau$ updates. The search is over either when the fitness if good enough, i.e., the maximum value exceeds a threshold $\upsilon$ or when a maximum number of iterations MAX has been reached. Let us denote by 2MES(x, $\sigma_0$, $\vartheta$, $\tau$, $\upsilon$, MAX, S, T) the 2MES procedure with the initial input vector $x = c_0$. The procedure computes the improved version of x, $x_{final}$, using the termination condition defined by the parameters $\upsilon$ and MAX, respectively.

Note that 2MES algorithm usually computes local optima and it is used to locally improve candidate solutions computed by global search procedures in hybrid or memetic approaches.

Firefly algorithm (FA) is a nature inspired optimization procedure, introduced in [30]. FA belongs to the class of swarm intelligence methods and it mimics the behavior of fireflies and their bioluminescent communication. The ideas underlying FA are that each firefly is attracted by the flashes emitted by all other fireflies, the attractiveness of an

individual is linked to the brightness of its flashes, and influenced by the light absorption and the law of light variations with distance.

In terms of image registration problem (10), the position of a firefly i corresponds to a candidate solution $c_i = (ca_i, cb_i, c\theta_i, cs_i)$, its light intensity being given by $fitness(c_i)$. For each pair of fireflies i and j, if j is brighter than i, that is $fitness(c_j^t) > fitness(c_i^t)$, then i is attracted by j, its position being updated based on the following equation:

$$c_i(t+1) = c_i(t) + \beta_j(r) \cdot \left(c_j(t) - c_i(t)\right) + \alpha \cdot \varepsilon \tag{27}$$

where $\alpha$ controls the randomness, $\varepsilon$ is randomly drawn from $U(0, 1)$, $\beta_j(r)$ is the attractiveness of j seen by i defined by:

$$\beta_j(r) = \beta_0 \cdot e^{-\gamma r^2} \tag{28}$$

where $r = \|x_j - x_i\|$ is the Euclidian distance between i and j. The constant $\beta_0$ is the brightness of any firefly at $r = 0$ and $\gamma$ represents the light absorption coefficient.

Usually, the update rule (25) is applied if the attractiveness of i in the new location $c_i(t+1)$ is higher than the attractiveness corresponding to the old position $c_i(t)$. The termination criterion of the FA is formulated in terms of number of iterations. Obviously, if the maximum value of the brightness function is known, the FA ends when the current best individual is good enough.

The image registration problem (10) has been solved using the fixed-size model of FA, where the population at time t, $t \geq 0$, has n individuals, $X^t = \{c_1^t, c_2^t, ..., c_n^t\}$, $c_i^t = (c_i^t(1), ..., c_i^t(4))$ and $c_i^t(k) \in [l(k), h(k)], k = 1, ..., 4$ [27]. Note that the search space $\prod_k [l(k), h(k)]$ used in the cited work is predefined, it does not change or adapt in any way depending on the image properties.

The initial population, $X_0$, is randomly generated according to the uniform probability distribution $U(l(k), h(k))$. We denote by cf a constant scale factor and let $U(a, b)$ be a draw from the uniform distribution on the interval [a, b]. The update rule introduced in [27] is given by:

$$c_i^{t+1}(k) = c_i^t(k) + \beta_{ij}(k) \cdot \left(c_j^t(k) - c_i^t(k)\right) + \frac{h(k) - l(k)}{\max\limits_k \left(h(k) - l(k)\right)} \cdot cf \cdot \exp\left(1 - fitness(c_j^t)\right) \cdot U(0,1) \tag{29}$$

In addition, a border reflection rule has been proposed to deal with unfeasibility:

$$\text{if } c_i^{t+1}(k) > h(k)$$

$$\text{then } c_i^{t+1}(k) = U\left(p \cdot l(k) + (1-p) \cdot h(k), b(k)\right)$$

$$\text{else} \tag{30}$$

$$\text{if } c_i^{t+1}(k) < l(k)$$

$$\text{then } c_i(k) = U\left(l(k), p \cdot l(k) + (1-p) \cdot h(k)\right)$$

where $p \in (0,1)$ and $U(a, b)$ represents a draw from uniform distribution on [a, b].

Note that in Attractiveness Formula (27) the quality of the attractor affects the randomness parameter. In case of high luminous intensity individual j, less randomness value is added. If the flashes emitted by the firefly j are weak then the perturbation grows.

### 3.3. Cluster-Based Memetic Registration

The proposed methodology is based on a core cluster-based memetic algorithm developed to register pairs of binary images. The global search procedure is directed by the variant of FA described in § 3.2, where the positions of fireflies belong to the domain



defined by (19). The fitness function defined by (22) implements Dice coefficient. Note that the maximum value of the fitness function is one.

The initial population is randomly generated and a small number of individuals are locally improved (a fixed percentage of population size). The local optimization is implemented by 2MES method provided in § 3.2. We denote by *nr* the number of individuals to be initially processed by 2MES.

The population-based optimization is an iterative process, at each iteration *t* the algorithm selecting pairs of distinct individuals $(c_i^t, c_j^t)$ and, if $\text{fitness}(c_j^t) > \text{fitness}(c_i^t)$, applies the update rule (27). If the brightness of the best individual in $X^{t+1}$ does not exceed the highest fitness value in $X^t$, then the local optimization procedure 2MES is used. The individuals selected for further improvements are computed based on clustered data. The candidate solutions in $X^{t+1}$ are grouped in k clusters using the Euclidian distance metric, one individual per cluster being chosen to be locally improved. The selection can be random or deterministic, for instance, one may choose either the centroid or the best candidate solution of each cluster.

In addition, the proposed hybridization between the population-based search and the local optimization procedure is designed to reduce the risk of premature convergence. Basically, two mechanisms are developed to deal with the situation of premature convergence. On one hand, at each iteration t, the number of clusters is set inverse proportional to the fitness value of the best individual in $X^{t+1}$, denoted by fitness (best). Let $k_0$ be the initial number of clusters, set as a small percentage of population size. We propose the computation rule:

$$k = k_0 \cdot \left\lfloor \frac{1}{\text{fitness(best)}} \right\rfloor \tag{31}$$

Moreover, the initial step size of 2MES procedure increases in case its consecutive iterations do not lead to quality improvement. The proposed update rule is given by:

$$\sigma = \frac{\sigma_0}{\text{fitness(best)}} \tag{32}$$

Note that $\text{fitness(best)} \leq 1$.

On the other hand, if the fitness is not improved over it2, it2 > it1, consecutive iterations, some new individuals are created to replace a set of randomly selected old ones. The newly created individuals are randomly generated using the uniform probability distribution, each one of them being improved next by 2MES procedure. We denote by $\text{NEW}(X^{t+1},\text{ind})$ the procedure that refreshes $X^{t+1}$ by adding individuals, as we explained above.

The search is over after NMAX iterations or when the best computed fitness value is above a threshold $\tau_{\text{stop}}$.

The detailed description of the proposed algorithm is provided below. We denote by S and T the sensed image and the target image, respectively. The parameters $\sigma_0, \vartheta, \tau_{ES}, \upsilon$ and MAX correspond to the 2MES procedure applied to improve the initial population and we denote by $\sigma'_0, \vartheta', \tau'_{ES}, \upsilon', \text{MAX}'$ the parameters of the local optimizer applied on clustered data. The parameters $\beta_0$ and $\gamma$ are specific to FA, according to §3.2 and let us denote by $X^t = \{c_1^t, c_2^t, ..., c_n^t\}$ the current population at the $t^{\text{th}}$ iteration. The variable counter counts the number of consecutive populations having the same best fitness value.

---

**Algorithm 1** Cluster-based memetic algorithm

1. **Inputs:** n, NMAX, $\tau_{\text{stop}}$, nr, $k_0$, ind, $\beta_0$, $\gamma$, cf, $\sigma_0$, $\vartheta$, $\tau_{ES}$,

    $\upsilon$, MAX, $\sigma'_0$, $\vartheta'$, $\tau'_{ES}$, $\upsilon'$, MAX', S, T

2. **Compute** $\mathcal{D}(S, T)$ according to (19)

3. t = 0; counter = 0

4. Compute $X^t$, the initial population according to Algorithm 2

5.  Evaluate the individuals in $X^t$ and compute best: $\text{fitness(best)} = \max_{x \in X^t} \text{fitness}(x)$

6.  **while** $t < \text{NMax}$ and $\text{fitness(best)} < \tau_{\text{stop}}$ **do**

7.      Apply an FA iteration, according to Algorithm 3

8.      Compute $\text{best}_c$: $\text{fitness(best}_c) = \max_{x \in X^{t+1}} \text{fitness}(x)$

9.      **if** $\text{fitness(best}_c) \leq \text{fitness(best)}$
10.         **if** counter == it1
11.             Update $\sigma'_0$ according to (30)
12.         **end if**
13.         **if** counter == it2 NEW($X^{t+1}$,ind)
14.         **end if**
15.         compute k according to (29)
16.         apply k-means to $X^{t+1}$ and obtain the clusters $C_1, \dots, C_k$
17.         **for** i = 1…k
18.             Select $x \in C_i$ –randomly / the centroid / the best candidate solution
19.             $x_{\text{new}}$ = 2MES($x, \sigma'_0, \vartheta', \tau'_{\text{ES}}, \upsilon', \text{MAX}', \text{S}, \text{T}$)
20.             Replace $x$ by $x_{\text{new}}$ in $X^{t+1}$
21.         **end for**
22.         Compute $\text{best}_c$: $\text{fitness(best}_c) = \max_{x \in X^{t+1}} \text{fitness}(x)$
23.         **if** $\text{fitness(best}_c) > \text{fitness(best)}$
24.             best = $\text{best}_c$; counter=0
25.         **else if** $\text{fitness(best}_c) < \text{fitness(best)}$
26.                 Randomly select $x \in X^{t+1}$
27.                 replace $x$ by best in$X^{t+1}$
28.             **else** counter = counter +1
29.             **end if**
30.         **end if**
31.     **end if**
32.     $t = t + 1$

33. **end while**
34. **Output: best,** $\widetilde{T}=S(g_{\text{best}})$ according to (21)

---

**Algorithm 2** Computation of the initial population

1.  **Inputs:** n, $\sigma_0$, $\vartheta$, $\tau_{\text{ES}}$, $\upsilon$, MAX, S, T
2.  Randomly generate an initial population $X^0 = \{c_1^0, c_2^0, \dots, c_n^0\}$
3.  **for** i = 1…nr
4.  Randomly select $x \in X^t$
5.  $x_{\text{new}} = 2\text{MES}(x, \sigma_0, \vartheta, \tau_{\text{ES}}, \upsilon, \text{MAX}, \text{S}, \text{T})$
6.  Replace $x$ by $x_{\text{new}}$ in $X^0$
7.  **end for**
8.  **Output:** $X^0$

---

**Algorithm 3** FA iteration

1.  **Inputs:** n, $X^t$, $\beta_0$, $\gamma$, cf, S, T
2.  **for** i = 1…n
3.      **for** j = 1…n
4.          **if** $\text{fitness}(c_j^t) > \text{fitness}(c_i^t)$

| | | |
|---|---|---|
| 5. | | Compute $c_i^{t+1}$ by moving firefly $i$ toward firefly $j$ using (27) |
| 6. | | Use the border reflection mechanism (28) to adjust $c_i^{t+1}$ |
| 7. | | Evaluate $c_i^{t+1}$ |
| 8. | **end if** | |
| 9. | **end for** | |
| 10. | **end for** | |
| 11. | **Output:** $X^{t+1}$ | |

### 3.4. Monochrome Image Registration

The method described by Algorithm 1 can also be applied, after a preprocessing stage, when monochrome images should be registered. Obviously, the main idea is to binarize the images by representing them using only the boundaries of their objects. However, depending on the complexity and quality of the analyzed images, further specific image processing techniques may be needed, as for instance image enhancement, de-blurring and noise removal. Alternatively, one can use edge detectors insensitive to noise and variations in illumination. Examples of such filters are reported in [31–33].

In the following we assume that the input images have already been processed such that a contour detection mechanism can be applied.

Let S and T be the $M \times N$ sized sensed and target image respectively and we assume that the rigid transformation is given by the parameters $(a, b, \theta, s)$. The proposed registration procedure consists of the following steps. First, a contraction mechanism, for example a scale transformation with supra-unitary factor, is applied. If we denote by $s_c$ the contraction factor, the images S and T are transformed according to:

$$S_p(x, y) = S\left(h(x, y)\right) \tag{33}$$

$$T_p(x, y) = T\left(h(x, y)\right) \tag{34}$$

where $1 \le x \le M, 1 \le y \le N$ and

$$h(x, y) = s_c \cdot \begin{bmatrix} x \\ y \end{bmatrix} \tag{35}$$

The main aim of the transforms (31) and (32) is to reduce the size of objects in the processed images and hence the complexity of search.

The next step is to represent $S_p$ and $T_p$ using only the contour of the objects belonging to the images. We denote by $OS_p$ and $OT_p$ the results of applying an edge detector to $S_p$ and $T_p$, respectively. Obviously, $OS_p$ is a perturbed version of $T_p$. We denote by $(ba, bb, b\theta, bs)$ the parameters of the corresponding perturbation. Using straightforward computation, we obtain:

$$a = s_c \cdot ba, b = s_c \cdot bb, \theta = b\theta, s = bs \tag{36}$$

Consequently, to compute the parameters $(a, b, \theta, s)$ we first apply the Algorithm 1 to obtain an approximation of $(ba, bb, b\theta, bs)$, and then use Equation (34).

### 3.5. Monochrome Image Registration in Case of Scaling on Multiple Dimensions

The proposed methodology can be extended to the case of more general perturbation models, in which each dimension is scaled with a specific stretching factor. The transformation is given by:

$$S(x, y) = T\left(f(x, y)\right) \tag{37}$$

where

$$f(x, y) = \begin{bmatrix} a \\ b \end{bmatrix} + \begin{bmatrix} s_x & 0 \\ 0 & s_y \end{bmatrix} \cdot R \cdot \begin{bmatrix} x \\ y \end{bmatrix} \tag{38}$$

for each $(x, y), 1 \le x \le M, 1 \le y \le N$. The scale matrix $s = \begin{bmatrix} s_x & 0 \\ 0 & s_y \end{bmatrix}$ is such that $s_x, s_y > 0$.

The search space boundaries can be computed similarly to (10) and (11). If $s_x \in (0, smax_x]$ and $s_y \in (0, smax_y]$, by straightforward computation we obtain,

$$Da = [minxT - smax_x \cdot \sqrt{2} \cdot maxS, maxxT + smax_x \cdot maxxS] \tag{39}$$

$$Db = [minyT - smax_y \cdot maxyS, maxyT + smax_y \cdot \sqrt{2} \cdot maxS] \tag{40}$$

Consequently, the proposed image alignment method aims to compute the parameters $(a, b, \theta, s_x, s_y)$ such that the relations (37) and (38) hold, where

$$(a, b, \theta, s_x, s_y) \in \mathcal{D}(S, T) = Da \times Db \times [-\pi, 0] \times (0, smax_x] \times (0, smax_y] \tag{41}$$

For each candidate solution $c = (ca, cb, c\theta, cs_x, cs_y)$, the fitness function measures the similarity between the target image T and the image $\widetilde{T}$,

$$\widetilde{T}(x, y) = S(g_c(x, y)) \tag{42}$$

$$g_c(x, y) = cR^T \cdot \begin{bmatrix} \dfrac{1}{cs_x} & 0 \\ 0 & \dfrac{1}{cs_y} \end{bmatrix} \cdot \left( \begin{bmatrix} x \\ y \end{bmatrix} - \begin{bmatrix} ca \\ cb \end{bmatrix} \right) \tag{43}$$

$$\text{fitness}(c) = \text{Similarity}(\widetilde{T}, T) \tag{44}$$

where $cR = \begin{bmatrix} \cos c\theta & -\sin c\theta \\ \sin c\theta & \cos c\theta \end{bmatrix}$.

The methodology described in § 3.4 can be applied to align images perturbed by (38) using the fitness function defined by (44).

## 4. Efficiency Measures

Let S be the sensed image and we denote by T the target. The images have the same size, $M \times N$. The accuracy of the registration method is measured through the success rate and using the similarity between T and the result of applying the alignment process on S, denoted by $\widetilde{T}$. We evaluate the similarity between T and $\widetilde{T}$ using two metrics commonly used in image processing, signal-to-noise-ratio (SNR) and peak-signal-to-noise ratio (PSNR), and two entropic measures, Shannon normalized mutual information defined by (4) and Tsallis normalized mutual information given by (9). The values $SNR(T, \widetilde{T})$ and $PSNR(T, \widetilde{T})$ are given by

$$SNR(T, \widetilde{T}) = 10 * \log_{10} \left[ \frac{\sum_{x=1}^{M} \sum_{y=1}^{N} (T(x, y))^2}{\sum_{x=1}^{M} \sum_{y=1}^{N} \left( T(x, y) - \widetilde{T}(x, y) \right)^2} \right] \tag{45}$$

$$PSNR(T, \widetilde{T}) = 10 * \log_{10} \left[ \frac{\max(T(x, y))^2}{\frac{1}{M \cdot N} \sum_{x=1}^{M} \sum_{y=1}^{N} \left( T(x, y) - \widetilde{T}(x, y) \right)^2} \right] \tag{46}$$

Note that the fitness function used to align pairs of images is also a similarity measure computed between binary images, Dice coefficient (1).

Due to the fact that the proposed methodology is of stochastic type, one way to evaluate its effectiveness is to compute the success rate, that is the percentage of runs that

led to the correct registration. From technical point of view, the result of applying Algorithm 1 to the pair (S, T) is correct if the obtained image $\tilde{T}$ corresponds to an individual best such that fitness(best)=Dice$(T, \tilde{T}) \geq \tau_{stop}$. Consequently, the success rate of the algorithm is given by:

$$SR(T, S) = \frac{NS}{NR} \cdot 100\% \tag{47}$$

where NS is the number of successful runs and NR is the total number of algorithm executions.

To evaluate the accuracy of the proposed method, we compute the mean values of the above-mentioned similarities. If we denote by $\tilde{T}_1, \ldots, \tilde{T}_{NR}$ the registered versions computed by the algorithm and $SIM \in \{DICE, NMI^S, NMI_\alpha^T, SNR, PSNR\}$, we obtain

$$MeanSIM(T, S) = \frac{\sum_{i=1}^{NR} SIM(T, \tilde{T}_i)}{NR} \tag{48}$$

The efficiency of the proposed method is experimentally evaluated by the runtime function. If we denote by $t_1, \ldots, t_{NR}$ the execution times consumed by Algorithm 1 to register the pair (S, T), the mean runtime value is computed by

$$MeanRT(T, S) = \frac{\sum_{i=1}^{NR} t_i}{NR} \tag{49}$$

From theoretical point of view, computational complexity may be evaluated in several ways. In our analysis we use the size of population (n) and maximum number of iterations corresponding to FA (NMAX) and 2MES (MAX, MAX', where MAX and MAX' have the same magnitude) to estimate the worst-case scenario. Since 2MES performs at most MAX iterations, the initial population generation algorithm (Algorithm 2) has a complexity of O(n*MAX). The complexity of Algorithm 1 is influenced by the k-means clustering algorithm. Since in our method the number of clusters linearly depends on the population size and the complexity of k-means is O(k*n*chromosome size), the resulting complexity of the clustering algorithm is O(n²). The 2MES performs at most, MAX' iterations and is called inside a loop depending on k, which means the complexity of this loop is O(n*MAX'). An iteration of FA algorithm (Algorithm 3) has a complexity of O(n²). Consequently, the complexity of the proposed method is O(NMAX*n*MAX). Note that in most cases n << MAX and NMAX and MAX have a similar magnitude, which leads to a quadratic complexity depending on the number of iterations.

## 5. Experimental Results and Discussion

To derive conclusions regarding the quality of the proposed approach a long series of test have been conducted on both binary and monochrome images. The results were obtained using the following configuration: processor Intel Core i7-10870H up to 5.0 GHz, 16 GB RAM DDR4, SSD 512 GB, NVIDIA GeForce GTX 1650Ti 4 GB GDDR6.

### 5.1. Binary Image Registration

Our tests have been conducted on a set of 16 binary images representing signatures, all having the same size $192 \times 192$ pixels. The images, denoted by $S_1, \ldots, S_{16}$, are perturbed by the rigid transformation (10) and (11) with various perturbation parameters. The rotation angle is between $-\pi$ and 0, while the scale factor was set in $[0.5, 1.5]$. The translation parameters are $a \in [-40, 10]$ and $b \in [40, 60]$. The rigid transformation parameters correspond to the working assumption that the perturbation process is totally reversible, that is the object pixels are completely encoded in the sensed images.

The search space is computed using (19). Note that the intervals $D_a$ and $D_b$ are significantly larger than $[-35, 10]$ and $[40, 60]$. For instance, in case of $S_1$, $D_a = [-402, 411]$ and $D_b = [-258, 579]$, while $a = -36$ and $b = 46$.

Since the perturbation process is totally reversible, the fitness threshold $\tau_{stop}$ is set close to the maximum value, one. In our test $\tau_{stop} = 0.9$. The rest of the input parameters are set as follows: n = 20, NMAX = 200, nr = 6, $k_0 = 4$, ind = 4, $\beta_0 = \gamma = 1$, cf = 2, $\sigma_0 =$ [7, 7, 0.3, 0.3], $\vartheta = \vartheta' = 0.85$, $\tau_{ES} = 20$, $\upsilon = 0.5$, MAX = 800, $\sigma'_0 = [3, 3, 0.02, 0.02]$, $\tau'_{ES} =$ 10 and MAX'=200.

The experimentally established results regarding the accuracy and the efficiency of Algorithm 1 are provided in Table 1. Note that the success rate is 100% for all pairs of images, NR = 700 and the SNR values are computed for images having the gray levels in {0, 1}. The computation is over when the maximum fitness value is at least 0.9.

**Table 1.** The results of applying Algorithm 1 in case of pairs of binary images.

| Input | MeanRT | MeanDice | MeanNMI$^S$ | MeanSNR | MeanPSNR |
|-------|--------|----------|-------------|---------|----------|
| $S_1$ | 11.45 | 0.92 | 0.80 | 20.96 | 70.19 |
| $S_2$ | 10.82 | 0.92 | 0.82 | 18.19 | 72.99 |
| $S_3$ | 10.05 | 0.92 | 0.81 | 20.44 | 70.73 |
| $S_4$ | 7.56 | 0.92 | 0.82 | 13.76 | 77.47 |
| $S_5$ | 12.43 | 0.91 | 0.81 | 11.93 | 79.39 |
| $S_6$ | 9.38 | 0.92 | 0.81 | 12.33 | 78.99 |
| $S_7$ | 8.28 | 0.92 | 0.81 | 11.57 | 79.73 |
| $S_8$ | 7.95 | 0.92 | 0.81 | 11.61 | 79.69 |
| $S_9$ | 10.30 | 0.92 | 0.81 | 13.13 | 78.09 |
| $S_{10}$ | 9.06 | 0.92 | 0.81 | 12.31 | 78.88 |
| $S_{11}$ | 9.15 | 0.92 | 0.82 | 12.9 | 78.31 |
| $S_{12}$ | 8.11 | 0.92 | 0.81 | 11.70 | 79.46 |
| $S_{13}$ | 11.97 | 0.92 | 0.81 | 12.24 | 78.94 |
| $S_{14}$ | 7.68 | 0.92 | 0.82 | 13.48 | 77.74 |
| $S_{15}$ | 9.93 | 0.92 | 0.78 | 9.13 | 81.81 |
| $S_{16}$ | 9.25 | 0.92 | 0.82 | 12.86 | 78.34 |

*5.2. Monochrome Image Registration*

In case of more complex, monochrome images, the assumption that the perturbation process is completely reversible is rather unrealistic. From the technical point of view, it means that the search procedure cannot manage to compute an individual with fitness 1, that is even when the rigid transformation parameters are correctly determined. Obviously, in such cases the threshold $\tau_{stop}$ should be set on lower values and the evaluation of accuracy should take into account the similarity between the aligned version of S and the initial image with the missing parts instead of the target T.

Our tests were conducted on images belonging to the well-known Yale Face Database [34, 35], which contains 165 greyscale images of 15 individuals, 11 images per subject/class. The results reported in Tables 2–4 refer to 30 images, two for each person, while the images displayed in Figures 1–10 correspond to two classes.

First, we provide a report regarding the new method proposed in § 3.4. Examples of target images, sensed ones and images obtained when the correct inverse rigid transformation is applied are presented in Figure 1 (Subject 5, the first image, corresponding to Row 5 in Tables 2–4) and Figure 2 respectively (Subject 14, the second image, corresponding to Row 29 in Tables 2–4).

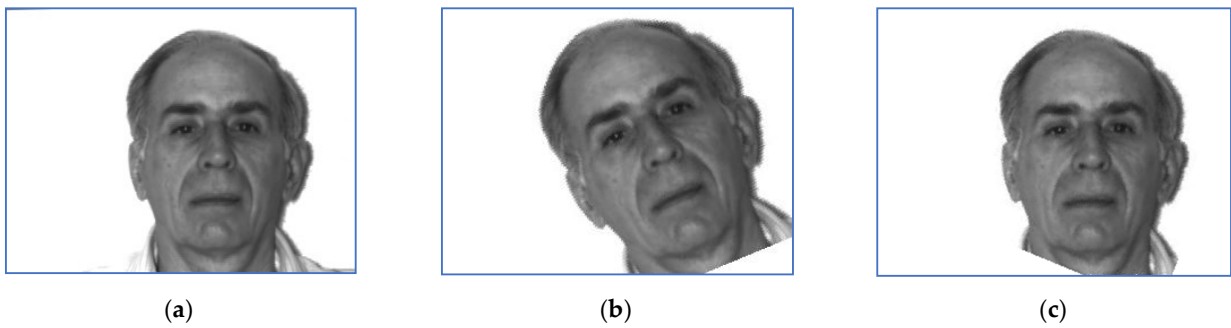

**Figure 1.** An image of Subject 5. (**a**) Target image, (**b**) sensed image, (**c**)correctly restored image.

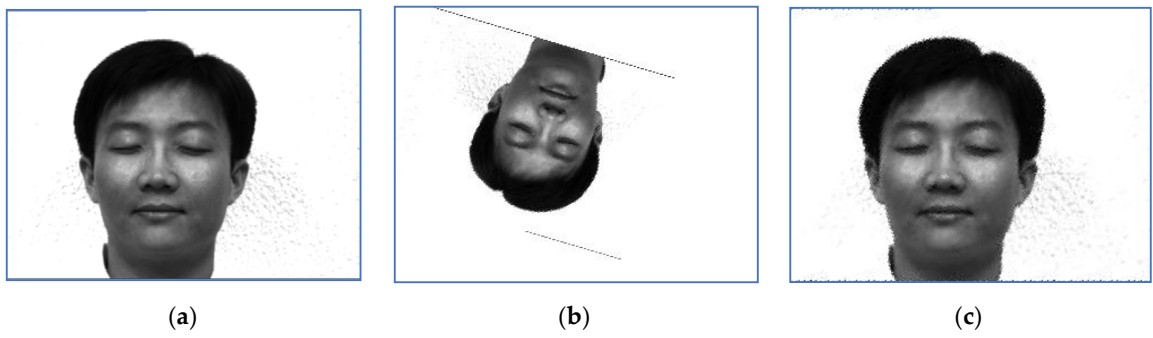

**Figure 2.** An image of Subject 14. (**a**) Target image, (**b**) sensed image, (**c**) correctly restored image.

The perturbation parameters are set as follows: the rotation angle is between $-\pi$ and 0, the scale factor was set in $[0.5, 1.5]$, and the translation parameters are $a \in [-130, 250]$ and $b \in [35, 420]$. For instance, the target image displayed in Figure 1 suffered a perturbation with the parameter vector $\left[-40, 70, -\frac{\pi}{8}, 0.9\right]$, while the one provided in Figure 2 was perturbed by the rigid transformation with $\left[230, 370, -\frac{\pi}{1.1}, 1.45\right]$.

Scaled and binarized versions of the pairs (S, T) are computed by the preprocessing step. The corresponding versions of the images displayed in Figures 1 and 2 are depicted in Figures 3 and 4. The scaling factor used for this step is 2.

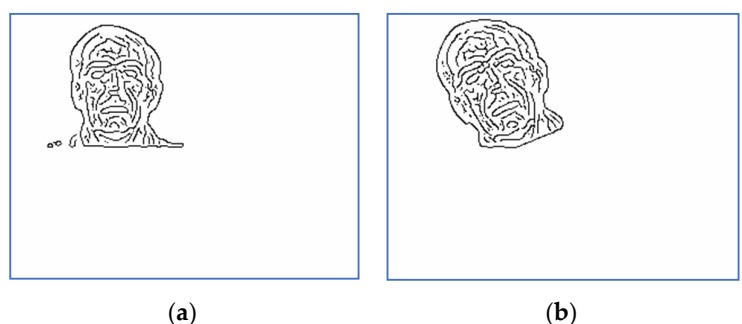

**Figure 3.** Scaled and binarized images for Subject 5: (**a**) target; (**b**) sensed.

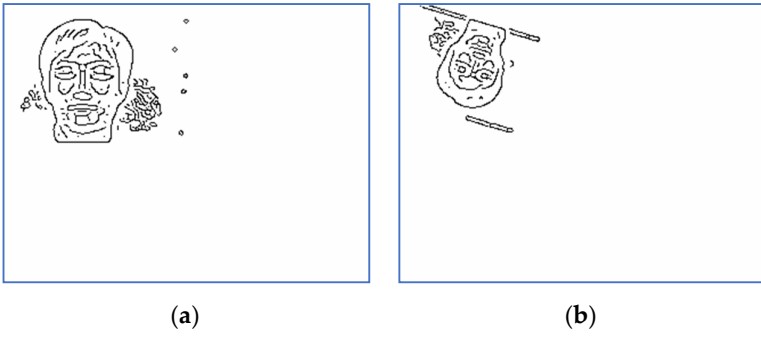

**Figure 4.** Scaled and binarized images for Subject 14: (**a**) target; (**b**) sensed.

Next, the Algorithm 1 is applied, where the inputs are the images computed by the preprocessing step. The search space is computed using (19). In our test $\tau_{stop} \in [0.5, 0.6]$. The rest of the input parameters are set as follows: n = 20, NMAX = 250, nr = 6, $k_0 = 4$, ind = 4 , $\beta_0 = \gamma = 1, cf = 2$ , $\sigma_0 = [12, 12, 0.3, 0.5]$ , $\vartheta = \vartheta' = 0.85$ , $\tau_{ES} = 20$ , $\upsilon = 0.3$ , MAX = 800, $\sigma'_0 = [7, 7, 0.03, 0.05]$, $\tau'_{ES} = 15$ and MAX'=240.

The results obtained by the proposed registration methodology are depicted in Figures 5 and 6.

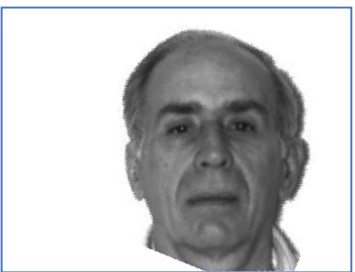

**Figure 5.** The restored image—Subject 5.

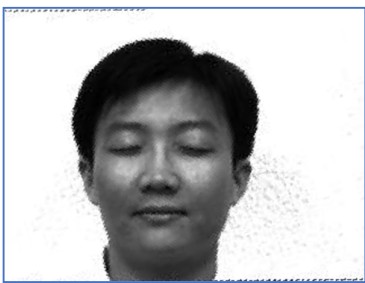

**Figure 6.** The restored image—Subject 14.

The accuracy and efficiency analysis are provided in Table 2. Note that the results concerning the similarity measures $SIM \in \{DICE, NMI^S, NMI^T_\alpha, SNR, PSNR\}$ are computed by

$$RSIM(\widetilde{T}) = \frac{MeanSIM(T', T)}{MeanSIM(\widetilde{T}, T')} \tag{48}$$

where T is the target, $\widetilde{T}$ is the restored version computed by the proposed methodology, and T' is the version obtained when the correct inverse rigid transformation is applied. In our tests, $\alpha = 1.3$ Obviously, the ideal value of the ratios defined by (48) is 1. However, possible larger values may be obtained, due to calculation and rounding errors.

The success rate of the proposed method is 100% for all the tested images, NR = 200 and the SNR values are computed for images having the gray levels in $\{0, \dots, 255\}$.

**Table 2.** The numerical results obtained by applying the proposed method.

| Image sample | MeanRT | RSNR | RNMI$^S$ | RNMI$^T_\alpha$ |
|:---:|:---:|:---:|:---:|:---:|
| 1 | 15.18996 | 0.84451 | 0.894946 | 0.976625 |
| 2 | 25.75912 | 0.930205 | 0.937661 | 0.987729 |
| 3 | 6.962834 | 0.758679 | 0.886308 | 0.968475 |
| 4 | 9.199112 | 0.818103 | 0.925562 | 0.980803 |
| 5 | 22.92094 | 0.984206 | 0.917182 | 0.982447 |
| 6 | 42.07603 | 0.755114 | 0.842885 | 0.960883 |
| 7 | 8.606454 | 0.942232 | 0.976699 | 0.993051 |
| 8 | 11.65704 | 0.973177 | 0.937504 | 0.983867 |
| 9 | 9.163015 | 0.860562 | 0.87384 | 0.974975 |
| 10 | 6.517512 | 0.87036 | 0.891611 | 0.980161 |
| 11 | 22.92636 | 0.820037 | 0.877844 | 0.973008 |
| 12 | 119.43 | 0.868211 | 0.917424 | 0.98466 |
| 13 | 18.72345 | 0.854931 | 0.95913 | 0.985814 |
| 14 | 16.16662 | 0.862684 | 0.962872 | 0.989826 |
| 15 | 39.514 | 0.961756 | 0.989651 | 0.995074 |
| 16 | 15.48423 | 0.947686 | 0.862359 | 0.973975 |
| 17 | 15.81341 | 0.965514 | 0.927276 | 0.983535 |
| 18 | 10.27014 | 0.955042 | 0.95205 | 0.98956 |
| 19 | 7.552746 | 0.864762 | 0.929975 | 0.971995 |
| 20 | 15.21946 | 0.902882 | 0.915916 | 0.976381 |
| 21 | 146.5344 | 0.845977 | 0.916805 | 0.985143 |
| 22 | 7.954541 | 0.728434 | 0.869876 | 0.964251 |
| 23 | 9.146519 | 0.861626 | 0.951103 | 0.986323 |
| 24 | 43.52619 | 0.904423 | 0.958607 | 0.984734 |
| 25 | 11.13669 | 0.816302 | 0.909504 | 0.966766 |
| 26 | 22.70944 | 0.87423 | 0.901217 | 0.97484 |
| 27 | 183.8919 | 0.924976 | 0.868897 | 0.97005 |
| 28 | 39.18939 | 0.791453 | 0.868468 | 0.961354 |
| 29 | 66.53452 | 0.875232 | 0.933425 | 0.975518 |
| 30 | 9.454251 | 0.968032 | 0.973915 | 0.992451 |

In order to analyze the registration capabilities of the proposed method, we experimentally compared it against two of the most commonly used align procedures in case of rigid transformation, namely one plus one evolutionary optimizer (EO) [21] and principal axes transform (PAT) [22]. Note that the function EO was tested with 100 different parameter settings per pair of images to establish the best alignment from the similarity ratio point of view (48), where SIM = NMI$^S$. The registered images using PAT method are displayed in Figures 7 and 8, while the results produced by EO are depicted in Figures 9 and 10.

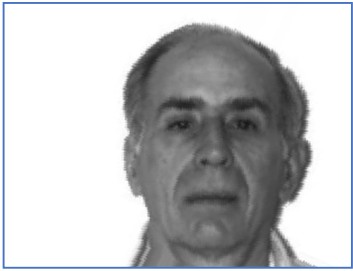

**Figure 7.** The restored image, PAT—Subject 5.

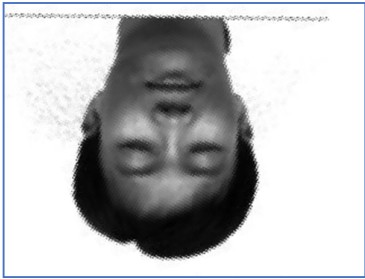

**Figure 8.** The restored image, PAT—Subject 14.

The numerical results are reported in Tables 3 and 4.

Note that PAT image alignment method has a widely known problem that in some cases produces results rotated 180 degrees along principal axes. In practice, this leads to some results being rotated upside-down. PAT stops at computing the aligned image and does not go further into analyzing if it is rotated or not, from a visual point of view. Some research [36] aims to correct such results by automatically assessing which of the two possible rotations represents the correct image. In case of images rotated to the left with large angles, PAT and EO may fail to provide the correct alignment. In such cases, the ratios values are significantly smaller than one. In case of PAT registration, the run time values vary between 4 and 6 s, while EO method consumes significantly more time due to the need to establish the appropriate input parameters.

**Table 3.** The numerical results obtained by applying the PAT method.

| Image sample | RSNR | $RNMI^S$ | $RNMI_\alpha^T$ |
|---|---|---|---|
| 1 | 0.803293 | 0.813467 | 0.933241 |
| 2 | 0.934574 | 0.886884 | 0.945906 |
| 3 | 0.295791 | 0.353257 | 0.696968 |
| 4 | 0.766659 | 0.843343 | 0.878612 |
| 5 | 0.905875 | 0.695198 | 0.943975 |
| 6 | 0.274874 | 0.372685 | 0.772999 |
| 7 | 0.840301 | 0.910059 | 0.950973 |
| 8 | 0.713365 | 0.527111 | 0.828832 |
| 9 | 0.626797 | 0.698461 | 0.841559 |
| 10 | 0.285219 | 0.351083 | 0.700893 |
| 11 | 0.251774 | 0.359619 | 0.739244 |
| 12 | 0.390197 | 0.418968 | 0.747504 |
| 13 | 0.80874 | 0.875054 | 0.925707 |
| 14 | 0.827627 | 0.892607 | 0.947912 |
| 15 | 0.366202 | 0.384068 | 0.732574 |
| 16 | 0.70648 | 0.550786 | 0.854011 |
| 17 | 0.68848 | 0.59478 | 0.841351 |
| 18 | 0.323253 | 0.393144 | 0.735975 |
| 19 | 0.34627 | 0.402099 | 0.734227 |
| 20 | 0.453927 | 0.423102 | 0.731206 |
| 21 | 0.191922 | 0.32247 | 0.810919 |
| 22 | 0.825042 | 0.910628 | 0.951726 |
| 23 | 0.851394 | 0.909486 | 0.949252 |
| 24 | 0.354395 | 0.424461 | 0.712203 |
| 25 | 0.27167 | 0.362291 | 0.697021 |
| 26 | 0.625461 | 0.788012 | 0.889674 |
| 27 | 0.406182 | 0.387472 | 0.767243 |

| | | | |
|---|---|---|---|
| 28 | 0.398996 | 0.443341 | 0.734644 |
| 29 | 0.337342 | 0.427012 | 0.748608 |
| 30 | 0.358785 | 0.423641 | 0.721053 |

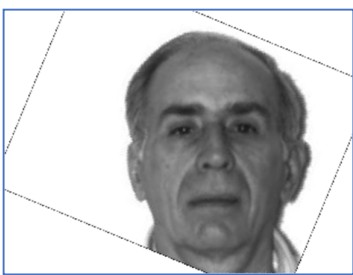

**Figure 9.** The restored image, EO—Subject 5.

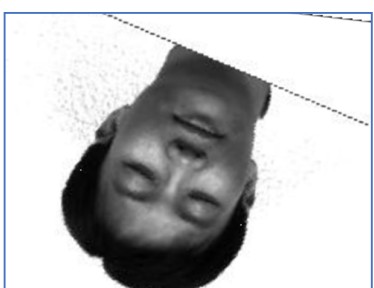

**Figure 10.** The restored image, EO—Subject 14.

**Table 4.** The numerical results obtained by applying the EO method.

| Image sample | RSNR | RNMI$^S$ | RNMI$_\alpha^T$ |
|---|---|---|---|
| 1 | 0.903736 | 1.028168 | 0.974588 |
| 2 | 0.59826 | 0.587583 | 0.829827 |
| 3 | 0.873805 | 1.011028 | 0.96525 |
| 4 | 0.881405 | 1.014717 | 0.967766 |
| 5 | 0.897092 | 1.039772 | 0.998585 |
| 6 | 0.327235 | 0.378149 | 0.699523 |
| 7 | 0.90252 | 1.030184 | 0.971119 |
| 8 | 0.961939 | 1.008078 | 0.976823 |
| 9 | 0.864434 | 1.013726 | 0.969337 |
| 10 | 0.518422 | 0.609677 | 0.861264 |
| 11 | 0.943244 | 1.027409 | 0.986622 |
| 12 | 0.33708 | 0.391352 | 0.7305 |
| 13 | 0.906637 | 1.011259 | 0.947328 |
| 14 | 0.997361 | 1.02143 | 0.959039 |
| 15 | 0.393484 | 0.435851 | 0.763053 |
| 16 | 0.907502 | 0.984902 | 0.965526 |
| 17 | 0.933475 | 1.002838 | 0.968684 |
| 18 | 0.247642 | 0.467926 | 0.840874 |
| 19 | 0.264723 | 0.436541 | 0.776755 |
| 20 | 0.781181 | 0.764776 | 0.911667 |
| 21 | 0.078978 | 0.30698 | 0.533648 |
| 22 | 0.917598 | 1.01558 | 0.961737 |
| 23 | 0.979704 | 1.030573 | 0.961777 |
| 24 | 0.318206 | 0.406862 | 0.601202 |

| 25 | 0.258663 | 0.367588 | 0.584295 |
| 26 | 0.258663 | 0.367588 | 0.584295 |
| 27 | 0.390247 | 0.407812 | 0.8087 |
| 28 | 0.391523 | 0.462968 | 0.745298 |
| 29 | 0.344073 | 0.496078 | 0.822182 |
| 30 | 0.886935 | 0.986715 | 0.955883 |

The results experimentally derived from a long series of tests lead to the conclusion that the proposed method outperforms PAT and EO from both accuracy points of view, informational and quantitative.

*5.3. Monochrome Image Registration in Case of Scaling on Multiple Dimensions*

The methodology introduced in §3.5 has been applied on images belonging to Yale Face Database perturbed according to (36). The results are as follows.

Examples of target images, sensed ones and images obtained when the correct inverse rigid transformation is applied are presented in Figure 11 (Subject 4, corresponding to Row 4 in Tables 5–7) and Figure 12 respectively (Subject 10, corresponding to Row 10 in Tables 5–7).

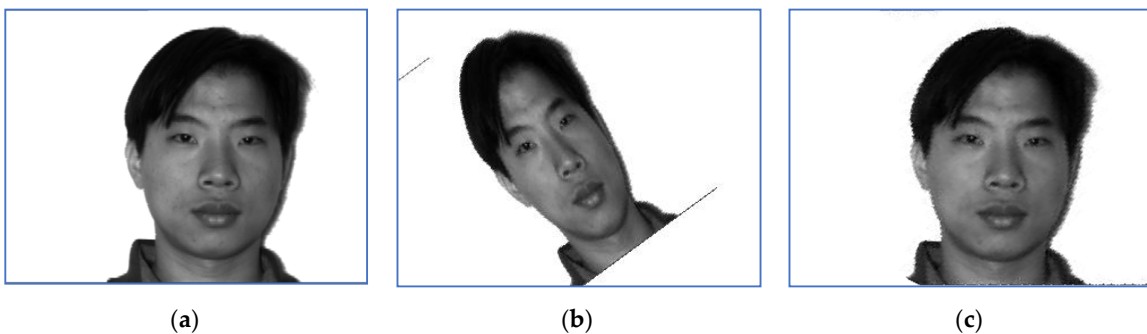

(**a**)   (**b**)   (**c**)

**Figure 11.** An image of Subject 4: (**a**) target image; (**b**) sensed image and (**c**) correctly restored image.

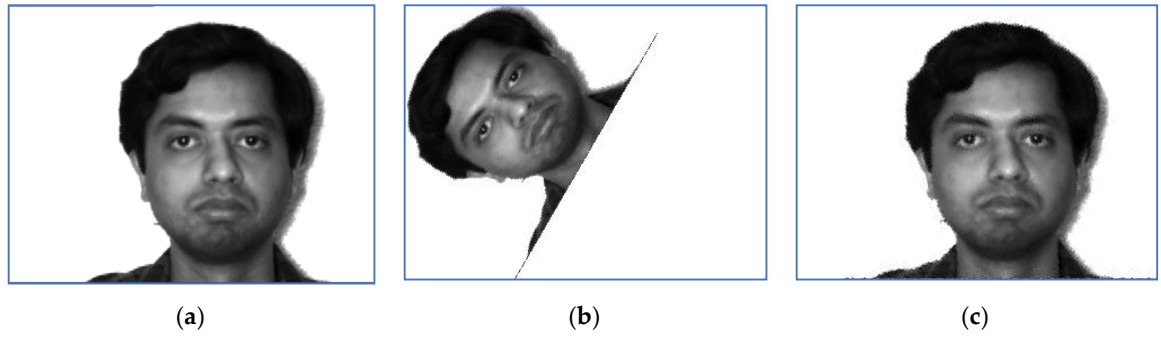

(**a**)   (**b**)   (**c**)

**Figure 12.** An image of Subject 10: (**a**) target image; (**b**) sensed image and (**c**) correctly restored image.

The perturbation parameters are set as follows: the rotation angle is between $-\pi$ and 0, the scale factors were set in $[0.5, 1.5]$, and the translation parameters are $a \in [-190, 245]$ and $b \in [35, 400]$. The target image displayed in Figure 11 suffered a perturbation with the parameter vector $\left[-50, 130, -\frac{\pi}{5}, 1, 1.4\right]$, while the one provided in Figure 12 was perturbed by the rigid transformation with $\left[-40, 220, -\frac{\pi}{3}, 1.37, 1.1\right]$.

The corresponding versions of scaled and binarized images displayed in Figure 11 and Figure 12 are depicted in Figures 13 and 14. The scaling factor used for this step is two.

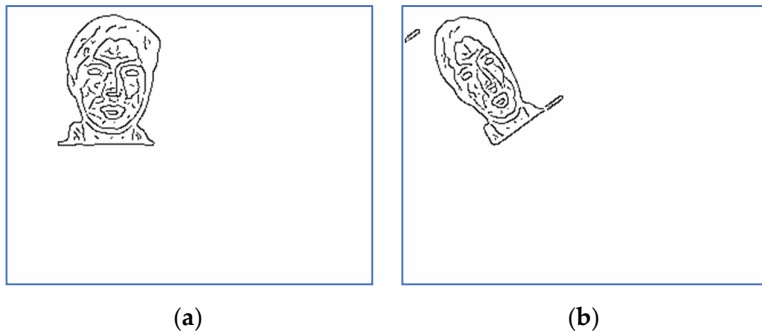

(**a**)　　　　　　　　　　　　　　(**b**)

**Figure 13.** Scaled and binarized images for Subject 4: (**a**) target; (**b**) sensed.

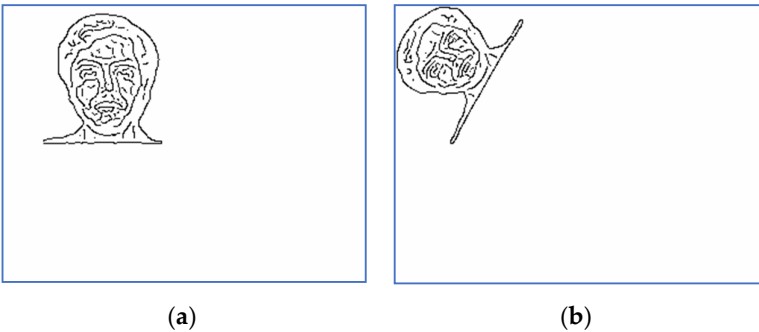

(**a**)　　　　　　　　　　　　　　(**b**)

**Figure 14.** Scaled and binarized images for Subject 10: (**a**) target; (**b**) sensed.

Next, the Algorithm 1 is applied, the input parameters being the same as those reported in §5.2. The results obtained by the proposed registration methodology are depicted in Figures 15 and 16.

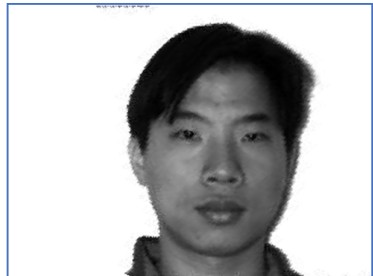

**Figure 15.** The restored image—Subject 4.

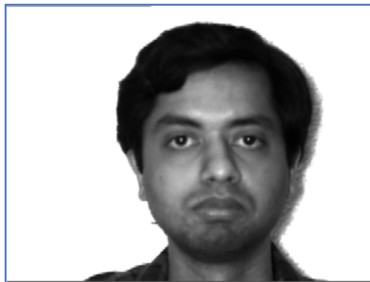

**Figure 16.** The restored image—Subject 10.

The accuracy and efficiency analysis are provided in Table 5. The results concerning the similarity measures $\text{SIM} \in \{\text{DICE}, \text{NMI}^S, \text{NMI}^T_\alpha, \text{SNR}, \text{PSNR}\}$ are computed by (40). The success rate of the proposed method is 100% for all the tested images, $\text{NR} = 50$ and the SNR values are computed for images having the gray levels in $\{0, \ldots, 255\}$.

**Table 5.** The numerical results obtained by applying the proposed method.

| Image Sample | MeanRT | RSNR | RNMI$^S$ | RNMI$^T_\alpha$ |
|---|---|---|---|---|
| 1 | 35.69 | 0.8391769 | 0.8791036 | 0.9670912 |
| 2 | 27.24 | 1.0049894 | 0.9447487 | 0.9942313 |
| 3 | 27.25 | 0.710606 | 0.8697491 | 0.9563255 |
| 4 | 45.82 | 0.8996928 | 0.9099942 | 0.9773764 |
| 5 | 21.8 | 0.8481722 | 0.868552 | 0.9696185 |
| 6 | 23.99 | 0.8540515 | 0.8962932 | 0.9770259 |
| 7 | 53.96 | 1.0575077 | 0.9456953 | 0.9982119 |
| 8 | 27.22 | 0.9761596 | 0.9007155 | 0.9778646 |
| 9 | 20.56 | 0.8486036 | 0.8658771 | 0.9687854 |
| 10 | 21.53 | 0.8431612 | 0.9251221 | 0.9780587 |
| 11 | 75.75 | 0.835015 | 0.8970438 | 0.9761927 |
| 12 | 89.93 | 0.8777121 | 0.8855052 | 0.9764733 |
| 13 | 43.54 | 0.8481775 | 0.9400461 | 0.9807903 |
| 14 | 55.53 | 0.8474774 | 0.9339562 | 0.9808082 |
| 15 | 188.34 | 0.8447351 | 0.906971 | 0.9555983 |
| 16 | 30.75 | 0.9905118 | 0.9202732 | 0.9832426 |
| 17 | 33.46 | 0.9655297 | 0.9767484 | 0.9851373 |
| 18 | 25.11 | 0.9837781 | 0.8992968 | 0.9727791 |
| 19 | 74.3 | 0.8561947 | 0.8659747 | 0.9607904 |
| 20 | 22.9 | 0.9264967 | 0.9124475 | 0.9794836 |
| 21 | 42.61 | 0.9677068 | 0.9124945 | 0.98168 |
| 22 | 20.47 | 0.7639291 | 0.8679323 | 0.9587384 |
| 23 | 31.92 | 0.7656339 | 0.8963442 | 0.9636461 |
| 24 | 128.13 | 0.7554826 | 0.869641 | 0.9579413 |
| 25 | 22.67 | 0.9521066 | 0.9514185 | 0.9883335 |
| 26 | 33.69 | 0.8187833 | 0.8784768 | 0.9676486 |
| 27 | 45.90 | 0.9023568 | 0.8995164 | 0.971906 |
| 28 | 41.43 | 0.8171742 | 0.9056656 | 0.9725819 |
| 29 | 40.83 | 0.7700865 | 0.9248933 | 0.972424 |
| 30 | 27.3 | 0.8174908 | 0.8947007 | 0.9685286 |

Note that PAT and EO algorithms lead to similar results as those reported in §5.2. (Figure 17–20). They fail to correctly perform the alignment from the rotation point of view, when the rotation angle is close to −180°. PAT may also misalign images in case of large magnitude distortions, mainly because the computation of the PAT solution involves arbitrary unitary matrices. Moreover, there are images with the same principal directions set and PAT cannot distinguish between them [37].

The numerical results are reported in Tables 6 and 7.

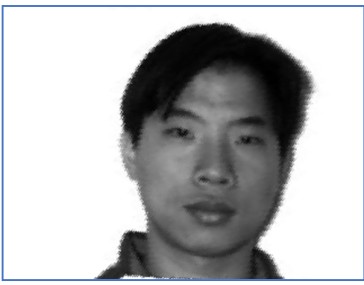

**Figure 17.** The restored image using PAT—Subject 4.

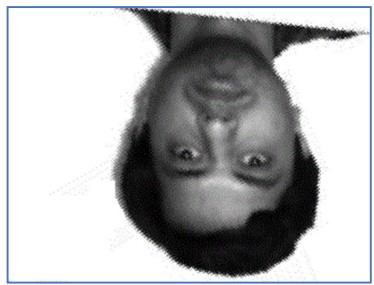

**Figure 18.** The restored image using PAT—Subject 10.

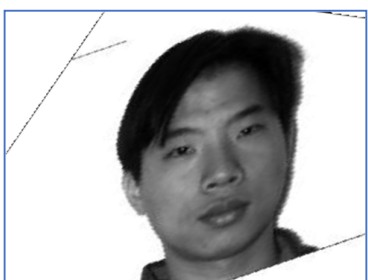

**Figure 19.** The restored image using EO—Subject 4.

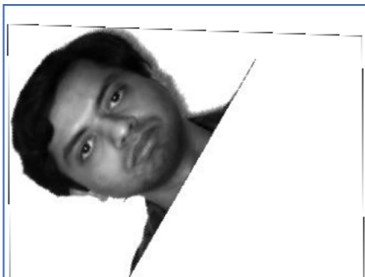

**Figure 20.** The restored image using EO—Subject 10.

**Table 6.** The numerical results obtained by applying the PAT method.

| Image Sample | RSNR | $RNMI^S$ | $RNMI^T_\alpha$ |
| --- | --- | --- | --- |
| 1 | 0.680357 | 0.671915 | 0.898062 |
| 2 | 0.730196 | 0.781502 | 0.860837 |
| 3 | 0.285552 | 0.34946 | 0.698664 |
| 4 | 0.702941 | 0.735872 | 0.912137 |
| 5 | 0.711288 | 0.669223 | 0.899995 |
| 6 | 0.28627 | 0.385159 | 0.772776 |
| 7 | 0.838095 | 0.948066 | 0.938333 |

| | | | |
|---|---|---|---|
| 8 | 0.786246 | 0.566011 | 0.860282 |
| 9 | 0.758941 | 0.755668 | 0.881690 |
| 10 | 0.282757 | 0.313553 | 0.656165 |
| 11 | 0.296401 | 0.372948 | 0.780554 |
| 12 | 0.410922 | 0.342484 | 0.622750 |
| 13 | 0.688558 | 0.739424 | 0.887327 |
| 14 | 0.388074 | 0.394377 | 0.731979 |
| 15 | 0.360452 | 0.378628 | 0.719361 |
| 16 | 0.858977 | 0.805554 | 0.941650 |
| 17 | 1.259501 | 1.187413 | 0.928381 |
| 18 | 0.37637 | 0.398532 | 0.784592 |
| 19 | 0.376674 | 0.371983 | 0.715186 |
| 20 | 0.472115 | 0.401741 | 0.744438 |
| 21 | 0.179625 | 0.315711 | 0.787056 |
| 22 | 0.850959 | 0.895747 | 0.940529 |
| 23 | 0.718642 | 0.834565 | 0.934975 |
| 24 | 0.335531 | 0.403226 | 0.699645 |
| 25 | 0.300216 | 0.348445 | 0.690366 |
| 26 | 0.354087 | 0.385384 | 0.759621 |
| 27 | 0.41668 | 0.373188 | 0.773804 |
| 28 | 0.391013 | 0.436276 | 0.727903 |
| 29 | 0.346801 | 0.431099 | 0.770062 |
| 30 | 0.510231 | 0.517476 | 0.729654 |

**Table 7.** The numerical results obtained by applying the EO method.

| Image Sample | RSNR | $RNMI^S$ | $RNMI_\alpha^T$ |
|---|---|---|---|
| 1 | 0.4840495 | 0.4851264 | 0.8291119 |
| 2 | 0.4560961 | 0.4942083 | 0.7928845 |
| 3 | 0.3180729 | 0.3866996 | 0.6517158 |
| 4 | 0.4961064 | 0.5416556 | 0.8193649 |
| 5 | 0.5940445 | 0.6214704 | 0.8908949 |
| 6 | 0.327612 | 0.3990796 | 0.7584589 |
| 7 | 0.5973359 | 0.6846458 | 0.8605331 |
| 8 | 0.8603518 | 0.7446469 | 0.9115472 |
| 9 | 0.5386513 | 0.5527723 | 0.8304493 |
| 10 | 0.1334781 | 0.1532915 | 0.2963478 |
| 11 | 0.4044389 | 0.4726133 | 0.8341134 |
| 12 | 0.2202632 | 0.1408288 | 0.167821 |
| 13 | 0.5098196 | 0.5599219 | 0.8161328 |
| 14 | 0.4358168 | 0.4987585 | 0.7843876 |
| 15 | 0.2878739 | 0.3921859 | 0.6948516 |
| 16 | 0.9984082 | 0.8356535 | 0.9701007 |
| 17 | 1.2488698 | 0.9357247 | 0.9713906 |
| 18 | 0.2557719 | 0.4326906 | 0.7551308 |
| 19 | 0.1200548 | 0.1649643 | 0.2182064 |
| 20 | 0.4760754 | 0.4876099 | 0.8062592 |
| 21 | 0.0116494 | 0.315598 | 0.5602149 |
| 22 | 0.5250537 | 0.5773189 | 0.8285102 |
| 23 | 0.4817668 | 0.5860995 | 0.8326726 |
| 24 | 0.2945408 | 0.4187963 | 0.6699495 |
| 25 | 0.3147539 | 0.385577 | 0.6504573 |

| 26 | 0.4207969 | 0.4808988 | 0.8230934 |
| 27 | 0.1889474 | 0.2075459 | 0.4060359 |
| 28 | 0.4167164 | 0.4874189 | 0.7887366 |
| 29 | 0.3523738 | 0.4888901 | 0.8267513 |
| 30 | 0.2477579 | 0.3808104 | 0.4784807 |

## 6. Conclusions

The work reported in this paper focuses on developing a novel and comprehensive methodology for image registration. The primary algorithm introduced in §3.3 addresses the problem of binary images, while its extended version deals with more complex, monochrome images. Note that the proposed approach could be further extended to colored images, the problem of representing the objects by their contour being solved either directly, using contour following algorithms or edge detectors, or after an additional procedure that computes the grey level versions of the inputs.

The main advantages of the proposed methodology are the comprehensiveness, due to the way we compute the search space, and the effectiveness, mainly due to the properties of the memetic firefly algorithm—ES approach, the embedded clustering procedure and the two mechanisms implemented to alleviate the risk of premature convergence.

The experimental results were derived based on very large number of tests and using various accuracy and efficiency measures. Both information-based similarity functions and quantitative measures, as for instance SNR and PSNR, were used to evaluate the effectiveness of our method and to compare it against two of the most commonly used align procedures in case of rigid transformation, EO algorithm and PAT method.

In case of binary images, the success rate is 100%, i.e., the target images are always identified by applying the inverse rigid transform on their corresponding sensed images, where the parameter vectors are computed by Algorithm 1. The recorded runtime values proved that the method is also efficient, especially being given its stochastic properties. The general methodology dealing with monochrome images also proved effective and efficient. Moreover, unlike PAT registration or EO, the proposed approach manages to correctly reverse the perturbation for all tested pair of images.

We conclude that the results are encouraging and entail future work toward extending this approach to more complex perturbation models as well as more advanced bio-inspired optimizations and evolutionary algorithms.

**Author Contributions:** Conceptualization, C.L.C. and C.R.U.; formal analysis, C.L.C. and C.R.U.; methodology, C.L.C.; software, C.L.C. and C.R.U.; supervision, C.L.C.; validation, C.L.C. and C.R.U.; writing—original draft, C.L.C.; writing—review and editing, C.L.C. and C.R.U.. All authors have read and agreed to the published version of the manuscript.

**Funding:** This research received no external funding.

**Conflicts of Interest:** The authors declare no conflicts of interest.

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
