# Peer review of "Cluster-Based Memetic Approach of Image Alignment"

_electronics, doi:10.3390/electronics10212606_

Round 1
Reviewer 1 Report
This paper presents a method for image alignment. The title of the article should be changed. Registration is the assembling of several images into a single integrated image.
Many experiments are presented for images of different scales and rotated images. It is worth checking the influence of noise on the results of the algorithms used. No analysis of the computational complexity of the presented algorithm has been presented.
Author Response
The authors thank the reviewer for the valuable suggestions.
Point 1: The title of the article should be changed. Registration is the assembling of several images into a single integrated image.
Response 1: We changed the title according to your suggestion. In some research works we found that the term alignment is also referred as registration, since the main purpose is the same.
Point 2: It is worth checking the influence of noise on the results of the algorithms used.
Response 2: Concerning the influence of noise, we added the following: depending on the complexity and quality of the analyzed images, further specific image processing techniques may be needed, as for instance image enhancement, de-blurring and noise removal. Alternatively, one can use edge detectors insensitive to noise and variations in illumination. Examples of such filters are reported in [31-33].
At this point we did not work on noisy images, but, as mentioned in the conclusions section, we intend to extend the research to more complex perturbation models.
Point 3: No analysis of the computational complexity of the presented algorithm has been presented.
Response 3: we added the following about the computational complexity:
From theoretical point of view, computational complexity may be evaluated in several ways. In our analysis we use the size of population (n) and maximum number of iterations corresponding to FA (NMAX) and 2M-ES (MAX, MAX’, where MAX and MAX’ have the same magnitude) to estimate the worst-case scenario. Since 2M-ES performs at most MAX iterations, the initial population generation algorithm (Algorithm 2) has a complexity of O(n*MAX). The complexity of Algorithm 1 is influenced by the k-means clustering algorithm. Since in our method the number of clusters linearly depends on the population size and the complexity of k-means is O(k*n*chromosome size), the resulting complexity of the clustering algorithm is O(n2). 2M-ES performs at most MAX’ iterations and is called inside a loop depending on k, which means the complexity of this loop is O(n*MAX’). An iteration of FA algorithm (Algorithm 3) has a complexity of O(n2). Consequently, the complexity of the proposed method is O(NMAX*n*MAX). Note that in most cases n<<MAX and NMAX and MAX have a similar magnitude, which leads to a quadratic complexity depending on the number of iterations.
Reviewer 2 Report
The manuscript describes a hybrid implementation of evolutionary algorithms to perform image registration. The method is sound but the problem is fairly trivial. The dataset used for deriving the results is fairly simple (mostly rotations).
Eq. 11 requires better formulation to include scaling on multiple dimensions. It is common to have optical distortions in images to be registered.
The manuscript does not detail the effect of different EA variables on accuracy and convergence time (e.g. population size, etc).
The authors should explain why the PAT method produces 180 rotated images. It sounds like a software bug due to use of `atan` instead of `atan2`.
Author Response
The authors thank the reviewer for the valuable suggestions.
Point 1: The method is sound but the problem is fairly trivial. The dataset used for deriving the results is fairly simple (mostly rotations). Eq. 11 requires better formulation to include scaling on multiple dimensions. It is common to have optical distortions in images to be registered.
Response 1: we have extended our method to more complex perturbations that involve rotation, translation and scaling with different factor on each axis. We are thankful for this idea because it confirmed the soundness of the proposed methodology. Consequently, new subsections were added to describe both the method and the experimental results: §3.5 and §5.3.
Point 2: The manuscript does not detail the effect of different EA variables on accuracy and convergence time (e.g. population size, etc.).
Response 2: we added the following paragraph regarding the influence of population size and number of iterations on the algorithm performances:
From theoretical point of view, computational complexity may be evaluated in several ways. In our analysis we use the size of population (n) and maximum number of iterations corresponding to FA (NMAX) and 2M-ES (MAX, MAX’, where MAX and MAX’ have the same magnitude) to estimate the worst-case scenario. Since 2M-ES performs at most MAX iterations, the initial population generation algorithm (Algorithm 2) has a complexity of O(n*MAX). The complexity of Algorithm 1 is influenced by the k-means clustering algorithm. Since in our method the number of clusters linearly depends on the population size and the complexity of k-means is O(k*n*chromosome size), the resulting complexity of the clustering algorithm is O(n2). 2M-ES performs at most MAX’ iterations and is called inside a loop depending on k, which means the complexity of this loop is O(n*MAX’). An iteration of FA algorithm (Algorithm 3) has a complexity of O(n2). Consequently, the complexity of the proposed method is O(NMAX*n*MAX). Note that in most cases n<<MAX and NMAX and MAX have a similar magnitude, which leads to a quadratic complexity depending on the number of iterations.
Point 3: The authors should explain why the PAT method produces 180 rotated images. It sounds like a software bug due to use of `atan` instead of `atan2`.
Response 3: PAT image alignment method has a widely known problem that in some cases produces results rotated 180 degrees along principal axes. In practice, this leads to some results being rotated upside-down.
The problem does not yield from a particular implementation of PAT (i.e. using a specific function instead of another – atan vs. atan2 in Matlab), but has a deeper mathematical root. PAT uses eigenvectors and eigenvectors are not unique. Eigenvectors are computed by calling a library function (eig in Matlab). Eigenvectors indicate the principal directions in Principal Component Analysis (PCA) and this means for the same direction two orientations are possible, both being equally mathematically correct, regardless of the visual orientation. Therefore, PAT may use a 180 degrees rotated orientation when aligning an image and produce an upside-down result, from a visual point of view.
PAT stops at computing the aligned image and does not go further into analyzing if it is rotated or not, from a visual point of view. Some research [35] aims to correct such results by automatically assessing which of the two possible rotations represents the correct image.
PAT may also misalign images in case of large magnitude distortions, mainly because the computation of the PAT solution involves arbitrary unitary matrices. Moreover, there are images with the same principal directions set and PAT cannot distinguish between them [36].
We have added the above-mentioned explanation in the article, in §5.2 and §5.3.
Reviewer 3 Report
Dear authors,
First of all, I would like to congratulate to the authors for this work. Absolutely, it is an interesting topic proposing a developing a novel and comprehensive methodology for image registration. Indeed, two algorithms are proposed obtaining an interesting and tested results. I consider that it is a great advance in this area that it will have a great improvement.
This paper is perfectly written, understandable, and well organized. I have learnt a lot with your proposal, and the literature review is excellent. The proposed solution is really well explained and a high quality of scientific impact. Moreover, figures and results are excellent.
The experimental results were derived based on very large number of tests and using various accuracy and efficiency measures, and the methodology and the implementation structure are excellent.
For this reason, I accept this version to be accepted for publish.
Best regards,
Author Response
The authors are very grateful to the reviewer for the appreciations. We thank the reviewer for the kind words and encouragement to continue researching this field.
Round 2
Reviewer 1 Report
I propose to change the title of chapter 3 - change the word registration to alignment.
Reviewer 2 Report
Authors addressed the requested comments.